# Faecal Immunochemical Test Impact on Prognosis of Colorectal Cancer Detected in Symptomatic Patients

**DOI:** 10.3390/diagnostics12041013

**Published:** 2022-04-17

**Authors:** Jesús Daniel Fernández de Castro, Franco Baiocchi Ureta, Raquel Fernández González, Noel Pin Vieito, Joaquín Cubiella Fernández

**Affiliations:** 1Department of Gastroenterology, Complexo Hospitalario Universitario de Ourense, 32005 Ourense, Spain; noel.pin.vieito@sergas.es (N.P.V.); joaquin.cubiella.fernandez@sergas.es (J.C.F.); 2Department of Gastroenterology, Hospital Universitario Lucus Augusti, 27003 Lugo, Spain; franco.baiocchi.ureta@sergas.es; 3Department of Internal Medicina, Complexo Hospitalario Universitario de Ourense, 32005 Ourense, Spain; raquel.fernandez.gonzalez3@sergas.es

**Keywords:** colorectal cancer, diagnosis, FIT, symptomatic

## Abstract

The use of the faecal immunochemical test (FIT) to stratify the risk of colorectal cancer (CRC) in symptomatic patients in primary healthcare enables improved referrals to colonoscopy. However, its effect on diagnostic delays or the prognosis of patients has been poorly evaluated in this setting. We performed a retrospective cohort study that included symptomatic patients with outpatient CRC diagnosis between 2009 and 2017. We identified whether FIT had been analysed between initial healthcare contact and diagnostic confirmation. We included 589 patients (male = 65%, 71.7 ± 11.6 years, TNM IV = 17.1%) in the analysis. FIT was performed in 411 (69.8%) patients with a positive result (≥10 µg/g of faeces) in 96.4% of the evaluated patients. The use of FIT was associated with increased diagnostic delay (yes = 159 ± 277 days, no = 111 ± 172 days; *p* = 0.01). At five years follow up, 193 (32.8%) patients died (151 due to CRC). Mean survival was not modified by the use of FIT or its result (not performed = 46.8 ± 1.5 months, FIT+ = 48.9 ± 1 months, FIT− = 45.6 ± 5.5 months; *p* = 0.5) in Kaplan–Meier analysis, and was confirmed later in multivariate Cox regression analysis. In conclusion, FIT determination in symptomatic patients in primary healthcare did not modify CRC prognosis.

## 1. Introduction

Colorectal cancer (CRC) is globally the third most common cancer and second leading cause of death related to cancer [1], which makes it one of the most important health challenges in the Western world. Its main prognostic factor is diagnostic stage [2], whereby it is vitally important to detect the tumour in the early stages of the disease. There are to strategies for this purpose: CRC populational screening and early diagnosis in symptomatic patients. The diagnostic process in symptomatic patients is complex. It is determined mainly by the high prevalence of gastrointestinal symptoms in the general population, usually related to benign pathology, and the high level of health resources involved in evaluating these patients. The total time this process takes is called diagnostic delay.

Faecal immunohistochemical tests (FIT) are the most broadly used diagnostic tool to screen for CRC in asymptomatic patients, and the reveal increased survival rate [3]. In the last few years, the use of FIT in the diagnosis of patients with abdominal symptoms from the scope of primary care has been evaluated. Recent meta-analysis that included 22 studies and more than 60,000 patients concluded that FIT increases the yield of colonoscopies, effectively stratifies the risk of CRC, and presents good sensitivity for diagnosis with an assumable number of false negatives [4].

In this sense, although there is strong evidence for its implantation in care circuits for individuals with gastrointestinal symptoms, we do not know the impact of FIT on prognosis in patients. Available evidence in this regard is limited and shows selection bias [5,6].

For this reason, we designed a study to determine whether performing FIT during the diagnostic process modifies the prognosis of CRC. In addition, we evaluated the effect of using FIT on diagnostic delay.

## 2. Materials and Methods

This is a retrospective, cohort, observational study. Data from patients previously collected for a study on the effect of diagnostic delay in symptomatic patients were used [7]. All patients had been diagnosed with CRC as an outpatient in the endoscopy unit of Ourense Teaching Hospital between 2009 and 2017. Patients older than 18 diagnosed by means of a colonoscopy requested by their primary care physician or by the hospital specialist physician, given the presence of symptoms suggesting CRC, were included. In our healthcare structure, all patients are initially evaluated by a primary care physician who may request FIT or refer to a hospital specialist physician at their discretion, and are able to request a colonoscopy if the patient complies with a pre-established indication [8].

Exclusion criteria were as follows: (1) absence of pathological anatomy compatible with adenocarcinoma; (2) asymptomatic patients: screening programme, opportunistic screening, monitoring of polyps or request because of family history; (3) hospital admission.

The used test was OC-Sensor (Eiken Chemical Co. Ltd., Tokyo, Japan). We identified whether FIT had been analysed between the first contact with the primary care physician and diagnostic confirmation. This time was quantified, and the attributable delay to the healthcare system was considered. For patients who underwent more than one test, only the first test performed was considered. For this study, three cohorts were defined: FIT not performed, FIT-positive (faecal Hb ≥ 10 µg/g of stool) and FIT-negative (Hb < 10 µg/g of stool).

Moreover, information was collected from demographic variables and variables in regard to the diagnostic process: FIT result, referral to hospital specialist doctor, date of request, and care level requesting the colonoscopy. In the event of referral to the hospital specialist doctor, referral and consultation dates were collected. Collected variables in regard to the tumour were: tumour location (rectum, distal colon/proximal to the splenic angle) and tumour stage (TNM 7th version [9]). CRC location was classified according to ICD-9 codes in proximal colon (153.0, 1, 4, 6) distal colon (153.2, 3, 7), and rectum (154, excluding 154.2 and 154.3). Advanced CRC was deemed to be Stages III and IV. Lastly, five-year follow-up was collected after diagnosis with the patient’s final status and cause of death if there were any.

First, descriptive analysis of obtained data was performed: quantitative variables are shown as means and standard deviations, while qualitative variables are shown as absolute numbers and frequencies. To determine whether there were any statistically significant relationships, the chi-squared test was used for qualitative variables. Student’s *t* test was used for the relationship between request for FIT and time delays, while single-factor ANOVA was used to evaluate the relationship between FIT outcome on time delays. We analysed whether there were differences in the probability of mortality (overall, attributable to CRC) between the three cohorts by means of the Kaplan–Meier method and log-rank test. We considered the first five years from the moment of diagnosis for Kaplan–Meier analysis. We performed multivariate Cox regression analysis with the intention of monitoring confounding factors, such as age, sex, or tumour stage. The association was shown as hazard ratio (HR) and 95% CI. Statistical programme IBM SPSS Statistics (Armonk, NY, USA: IBM Corp) was used for statistical analysis.

## 3. Results

### 3.1. Data Description 

Over the analysed period, 3862 patients with colorectal lesions were referred to the gastrointestinal oncology consultation from the endoscopy unit to complete the study. A total of 589 patients were included (65% men, mean age 71.7 ± 11.6 years), who complied with the inclusion criteria and from whom all the data could be obtained. The total number of patients with CRC who underwent FIT was 411 (69.8%); of these, 396 (96.4%) were positive (faecal Hb ≥ 10 µg/g of stool). Table 1 shows the sample’s characteristics. 

### 3.2. Variables Related to Performing FIT

We determined which variables were associated with performing FIT during the diagnostic process. Performing the test was not associated with sex (*p* = 0.2) or age (*p* = 0.85). It was more likely to find FIT performed in patients not referred to a hospital specialist (79.8% not referred vs 65.3% referred; *p* < 0.01). Patients who had undergone FIT presented a longer total time compared to those who had not undergone FIT (159 ± 277 days versus 111 ± 172 days; *p* = 0.01). However, performing FIT did not significantly increase time to evaluation in the hospital specialist (83.9 ± 139.8 days vs 81.9 ± 111 days; *p* = 0.98). However, in patients with proximal CRC, a significantly higher number of FIT was performed (proximal colon: 81.7%, distal colon: 70.2% and rectum: 62.4%; *p* < 0.001). Regarding prognosis, we did not detect differences in the frequency of performing FIT according to tumour stage (TNM IV: 66.3%, TNM III: 68.1%, TNM II: 74.1%, TNM I: 71.8%; *p* = 0.5).

### 3.3. Variables Related to FIT Outcome

Table 2 shows the distribution of variables among the three study cohorts (FIT not performed, positive, negative). Significant differences were observed in regard to referral to the hospital specialist. The proportion of referred patients with FIT+ was lower than that of the rest of the groups (FIT+: 64.8%, FIT−: 92.9%, FIT not performed: 80.1%; *p* < 0.001). For FIT+ patients, colonoscopy was more commonly requested from primary care (FIT+: 50.8%, FIT−: 28.6%, FIT not performed: 35.1%; *p* = 0.01). Lastly, diagnostic delay showed statistically significant differences. These were observed to be higher for FIT-negative patients (FIT+: 154.97 ± 267.18 days, FIT−: 273.07 ± 469.05 days, FIT not performed: 111.20 ± 171.84 days; *p* = 0.02).

### 3.4. Variables Related to CRC

FIT sensitivity was not modified by tumour location (rectum: 96.7%, distal colon: 98.6%, proximal colon: 93.1%, *p* = 0.06) or tumour stage (I: 98.3%, II: 95.3%, III: 96.0%, IV: 97.0%; *p* = 0.7). However, a significantly higher proportion of CRC was detected, located in the proximal colon among those individuals with FIT− (FIT+: 27.3%, FIT−: 53.3%, FIT not performed: 14.6%; *p* < 0.001). Table 3 shows the results of variables related to tumour and prognosis. As shown in this table, we did not detect an association with tumour stage, both metastatic (TNM IV) and advanced (TNM III-IV). Performing FIT and its outcome did not impact overall mortality at five years or mortality attributed to CRC.

Regarding analysis of survival, we did not detect differences in the probability of overall mortality or associated with CRC both for mortality analysis at five years (Table 3) and for Kaplan–Meier analysis. Figure 1 shows the survival curve by CRC. Mean survival considering only death because of CRC was: positive FIT: 51.3 ± 1 months, FIT negative: 44.6 ± 5.9 months, FIT not performed: 49.0 ± 1.4 months (*p =* 0.1). Mean overall survival was FIT+: 48.9 ± 1 months, FIT−: 45.6 ± 5.5 months, FIT not performed = 46.8 ± 1.5 months (*p* = 0.5). Lastly, for multivariate Cox regression, variables independently associated with mortality associated with CRC were age (HR: 1.03, 95% CI 1.02–1.05), tumour stage (TNM III, HR: 9.14, 95% CI 2.22–37.65; TNM IV, HR: 72.88, 95% CI 17.82–297.99), and the request for colonoscopy from primary care (HR: 0.6, 95% CI 0.42–0.86).

## 4. Discussion

### 4.1. Summary of Findings

We analysed in a broad sample of patients with outpatient diagnosis of CRC the prognostic impact of performing FIT at five years. FIT does not affect prognosis: no relationship with advanced stages or survival was observed. We confirmed high diagnostic sensitivity regardless of location and stage at diagnosis. Moreover, it appears to modify diagnostic circuits, and the patient’s evaluation and additional test requested can be optimised.

### 4.2. Similarities and Differences with Other Studies

Despite current high-quality evidence to justify the use of FIT in primary care, there is limited evidence over its prognostic implication, given that there are few studies with this aim. Banaszkiewicz et al. included a 10-year follow up of a sample of 10,000 patients diagnosed from specialised care. Colonoscopy or FIT was requested according to the severity of symptoms [5]. The outcome was the identification of better prognosis in patients from whom FIT was requested. A second study by Gutierrez-Stampa et al. analysed the existence of a positive result for FIT the year prior to CRC diagnosis in 1527 patients, associating better stage and survival at three years than those with a negative result or not performing the test [6]. Despite its large sample, it lacks having confirmed the existence of symptoms in patients. In this same study, FIT was requested in a mere 20.7% of patients, an appreciable difference in regard to our 69.8%, which, despite being different health systems, coincide in the same country and on similar data collection dates, which suggests that the acceptance of FIT by clinicians may be highly variable across regions.

In our study, stage at diagnosis did not impact FIT outcome. In meta-analysis performed by Niedermaier et al. [10], which included diagnostic studies in symptomatic patients but also in screening, much lower sensitivity was observed in Stage I compared to the rest. We did not observe this in our study with a clearly smaller sample. However, sensitivities by stage in our study were higher overall.

The absence of prognostic relationship between diagnostic delay and stage of presentation of the CRC in symptomatic patients is a consistent finding with previous studies and meta-analysis [11,12], in which patients with lower diagnostic delays tended to present more advanced stages at diagnosis and thereby lower survival. In any case, previous studies already highlighted the complex relationship between delays and prognosis; the type of symptom at presentation is critical [13]. The most urgent studies in patients with more severe symptoms are related to worse prognosis for a more advanced disease, while in patients with more nonspecific symptoms, which usually condition longer study times, delay and prognosis are not as clearly related.

### 4.3. Strengths and Weaknesses

Our study is one of the few to explore prognosis of FT in a daily use scope as a diagnostic tool beyond screening for CRC. Our series is well-documented, having performed an indepth follow-up of patients and being able to confirm the existence of clinical symptoms compatible with CRC in all of them. Moreover, a standard work circuit was carried out, which enables extrapolating to other health systems that present a similar protocol. The sensitivity attained in our study was a little better than that of other studies [14]. This confirms once again the correct performance of FIT under real-life conditions.

However, we recognise various limitations in our study. First, this is a retrospective and single-centre study that may be impacted by the diagnostic habits of a primary care regional area. However, to obtain a solely outpatient sample, we excluded those patients hospitalised during their diagnostic process. However, recent studies observed that up to 40% of patients diagnosed outside a screening programme are hospitalised at some time in the study, which could limit generalising our conclusions [15]. In accordance with our protocols to request colonoscopy [8], the threshold considered by the primary care physician was 20 µg Hb/g. However, we decided to use the threshold of 10 µg Hb/g for analysis, as this is what currently predominates in clinical guidelines for symptomatic patients (NICE Guidelines NG12 [16]). This would have included positive FIT for patients treated as negative. In our study, two symptoms of nonabdominal origin were not deemed to be exclusive: constitutional syndrome and anaemia (ferropoenic or other), which had not been included in all previous studies [14]. The symptoms not collected as variables for the analysis and the comorbidity of patients, two variables that could impact the colonoscopy time or the prognosis of patients. On the other hand, we could have performed different survival analysis in order to control competing events, such as a Cox method evaluating the cause specific hazard. However, from our perspective, performing different statistical analysis does not change the final result that we present in the manuscript. We reached the same result in the different analyses that we performed: CRC stage at diagnosis, five-year mortality both overall and related to CRC, probability of survival, and the multivariable analysis to control confounding variables. Lastly, the sample, despite being large overall, was low in the FIT-negative group, which could turn out to be insufficient to detect significant differences.

### 4.4. Implications for Physicians and Managers

First, our results suggest that tumour location appears to affect the request for FIT. Patients with CRC proximal to the splenic angle have more possibilities of performing FIT. The difference in clinical presentations may have been one of the causes that could impact this behaviour of the primary care physician. Recent studies confirm that left-sided tumours have a more symptomatic presentation (haematochezia, rectal symptoms) and this may mean it is not as necessary to request further tests that target the origin of the colon symptoms [17]. However, the most common right CRC symptoms are latent or nonspecific (such as abdominal pain, ferropoenic anaemia or altered bowel habit) which most commonly have nontumour aetiology. The reference symptom was not collected in our study as this association was not expected.

Another relevant finding was that half of the false negatives presented a tumour proximal to the splenic angle. Previous studies in patients within the scope of screening had already highlighted the lower sensitivity of FIT to detect advanced lesions in the right colon [18]. This should be explored in subsequent studies and, if confirmed, should be known and borne in mind by requesting physicians. In any case, in our study, the number of false negatives only conditioned an increase in delay with no impact on survival.

Both the request for FIT and referral to a hospital specialist are expected reasons for an increase in delay in regard to the direct request for colonoscopy. This is despite the fact that FIT increases total delay and not time in the initial consultation with a hospital specialist or time to perform the colonoscopy requested by a hospital specialist. In general, these results suggest two main ways to manage by the primary care physician. Patients whose primary care physician requests for FIT are also more likely to request a colonoscopy, while patients without FIT are commonly referred to a hospital specialist. This appears to suggest that primary care physicians with a more independent profile support their suspicions with FIT, previously upon requesting an invasive test that entails risks. Performing an FIT could have delayed the request for colonoscopy in this context. However, if we wish to avoid unnecessary delays and waiting lists, it is more suitable to facilitate performing FIT and possible subsequent colonoscopy by the same primary care physician, especially for those routine cases that would not benefit from evaluation by a gastroenterologist.

Despite being a simple test available all over the world, FIT is not used routinely beyond screening because of the fear of false negatives, which renders a safety network necessary, and the hypothetical risk that could be entailed by an unmanageable number of referrals to colonoscopy as a consequence of false positives and delaying performing all colonoscopies overall [19]. However, it should be considered that a primary care physician’s alternatives to FIT are a request for the colonoscopy itself (whereby FIT would act as a filter for patients with a higher risk of CRC) or risk delaying the diagnosis with the clinical monitoring of symptoms. In this sense, our study did not show that this tool impacts the survival of patients, which should be the key point to consider in this context.

Lastly, our study’s results open up a series of questions that should be explored in subsequent studies. Among them, variables that would contribute to the request and result of the FIT (especially false negatives), and the type of symptom or location of the CRC.

## 5. Conclusions

In this retrospective cohort study, the use of FIT in symptomatic patients evaluated in primary healthcare does not modify the CRC prognosis, considering both diagnostic stage and five-year survival. On the other hand, diagnostic delays are modified when FIT is used to stratify risk, increasing delays.

## Figures and Tables

**Figure 1 diagnostics-12-01013-f001:**
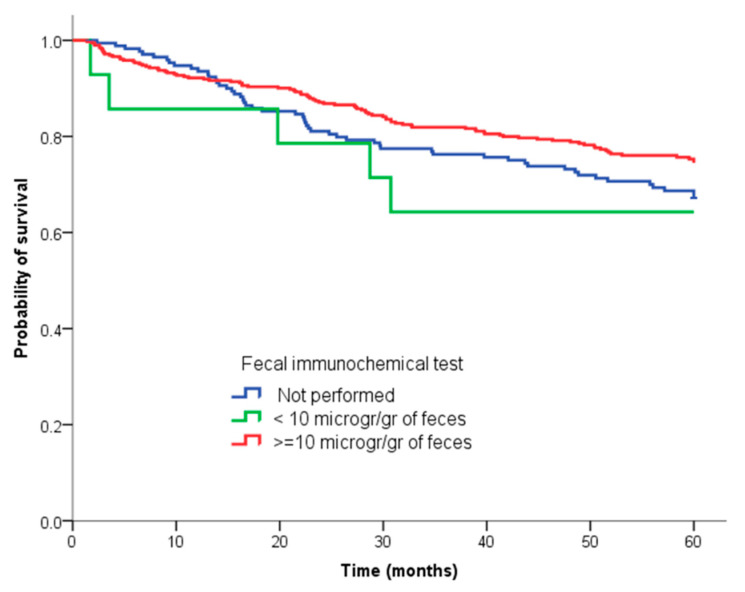
Survival curve of three cohorts calculated with Kaplan–Meier method.

**Table 1 diagnostics-12-01013-t001:** Description of patients included in analysis.

	Entire Cohort (589)
Age (years)	71.7 ± 11.6
Gender (males), n (%)	383 (65)
FIT performed, n (%)	411 (69.8)
FIT ≥ 10 µg/g faeces n (%)	396 (96.4)
Referral to specialist, n (%)	413 (70.2)
Colonoscopy from GP n (%)	256 (43.4)
Diagnostic delay (days)	145 ± 251
CRC location, n (%)	
Rectum	242 (41.1)
Left colon	205 (34.8)
Right colon	142 (24.1)
TNM, n (%)	
Stage I	85 (14.4)
Stage II	143 (24.3)
Stage III	260 (44.1)
Stage IV	101 (17.1)
Follow-up (months)	47.1 ± 18.9
Death, all causes, n (%)	193 (32.8)
Death from CRC, n (%)	151 (26.2)

CRC, colorectal cancer; FIT, faecal immunochemical test; GP, general practitioner. Qualitative variables are expressed as number and percentage. Quantitative variables are expressed as mean and standard deviation.

**Table 2 diagnostics-12-01013-t002:** Characteristics of cohorts.

	FIT Not Performed (n = 178)	FIT < 10 µg/g (n = 15)	FIT ≥ 10 µg/g (n = 396)	*p* ^1^
Gender (male)	109 (61.2%)	12 (80%)	262 (66.2%)	0.2
Age (years)	70.88 ± 11.54	76.5 ± 7.15	71.94 ± 11.78	0.1
Colonoscopy request from Primary Healthcare (yes)	60 (35.1)	4 (28.6)	192 (50.8)	<0.001
Referral to Secondary Healthcare (yes)	137 (80.1)	13 (92.9)	245 (64.8)	<0.001
Diagnostic delay (days)	111.20 ± 171.84	273.07 ± 469.05	154.97 ± 267.18	0.02

^1^ Significance in univariate analysis using chi-squared test for qualitative variables and ANOVA test for quantitative variables. Quantitative variables expressed as mean and standard deviation. Qualitative variables expressed as number and percentage.

**Table 3 diagnostics-12-01013-t003:** Effect of faecal immunochemical test on colorectal cancer location, stage, and prognosis.

	FIT Not Performed (n = 178)	FIT < 10 µg/g (n = 15)	FIT ≥ 10 µg/g (n = 396)	*p* ^1^
CRC location, n (%) Rectum Distal to splenic angle Proximal to splenic angle	91 (51.1%) 61 (34.3%) 26 (14.6%)	5 (33.3%) 2 (13.3%) 8 (53.3%)	146 (36.9%) 142 (35.9%) 108 (27.3%)	<0.001
TNM, n (%) Stage I Stage II Stage III Stage IV	24 (13.5%) 37 (20.8%) 83 (46.6%) 34 (19.1%)	1 (6.7%) 5 (33.3%) 7 (46.7%) 2 (13.3%)	60 (15.2%) 101 (25.2%) 170 (42.9%) 65 (16.4%)	0.7
Five-year mortality, n (%)	64 (35.9%)	5 (33.3%)	124 (31.3%)	0.5
Five-year CRC mortality, n (%)	54 (30.9%)	5 (33.3%)	92 (23.7%)	0.1

^1^ Significance in univariate analysis using chi-squared test for qualitative variables. CRC, colorectal cancer. Qualitative variables expressed as number and percentage. Quantitative variables expressed as mean and standard deviation.

## Data Availability

Data presented in this study are available on request from the corresponding author.

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
