# Peer review of "Faecal Immunochemical Test Impact on Prognosis of Colorectal Cancer Detected in Symptomatic Patients"

_diagnostics, 2022, doi:10.3390/diagnostics12041013_

Round 1
Reviewer 1 Report
Thank you very much for the opportunity to review this manuscript (ID: diagnostics-1649424). I listed my specific considerations below
ABSTRACT SECTION Lines 20-21
The submitted conclusion does not fully meet the aim of the study notified under lines 49-51.
MATERIALS AND METHODS SECTION
In this section, I did not find the characteristics of the study group. I found it in RESULTS SECTION lines: 101-106. Additionally, in my opinion, METHODS SECTON has a number of shortcomings. I suggest you give the detailed cancer codes according to the analyzed locations (eg ICD10 or ICDO3). Line 91, what different variables do the authors mean? Line 96, the impact of what confounding factors the authors feared? Is the Kaplan-Meier method used to assess mortality? In the description of this method, it is also worth mentioning which moment was considered "start" and which was considered "end". However, this is not the biggest problem. In the presented study, the authors had extremely valuable information that they did not use or misused. I mean cause of death. In the case of having this important information, the competing risk method should be used in time-to-event analyzes. Analyzing CRC specific and non-specific deaths separately is, in my opinion, a methodological error. The distribution of cancer stage as a factor associated with the cause of death was likely different between the two groups. In my opinion, cause specific hazard or subdistribution hazard ratio methods should have been used. In line 151 the authors write: "FIT sensitivity was not modified by tumor location ....". By what method was this sensitivity estimated? I suggest that you clarify this in the METHODS SECTION.
RESULTS SECTION
Line 108. Do demographic variables not characterize the study group? Lines 108-114. Part of the paragraph provides the exact information contained in table1. Table 1. Why is sd for age shown with a different accuracy than mean? Lines 129-131. How do the authors know that the proximal CRC was significantly different from other locations? Has one of the multiple comparison methods been used? If so, it is worth mentioning in the METHODS SECTION.
CONCLUSIONS SECTION
As in ABSTRACT SECTION, the presented conclusion does not confirm the full achievement of the study aim.
EVALUATION
An interesting study, however, has methodological flaws.
Best regards,
Reviewer
Reviewer 2 Report
Title change suggest: Faecal immunochemical test impact on prognosis of colorectal cancer in symptomatic patients.
Line 10, improved NOT improving
Line 16; did FIT increase or was it associated with an increased delay?
It is difficult to be sure of completeness of data in such a retrospective study.
FIT is all about identifying a group who may have a diagnosis and not about prognosis, so the clinical relevance of this study will not change the use of FIT.
Round 2
Reviewer 1 Report
I have read the manuscript again (ID: diagnostics-1649424). Thank you for responding to most of my comments. I have presented the remaining doubts below.
Applies to response 3: Cancer codes are a good and widely used classification now, and not in the future. The code also includes information about the topography of the tumor. Any other classification, eg anatomical, complements these codes. It is most often used due to differences in prognosis. If there is no histopathological confirmation and no established code, eg ICD, on what basis was CRC confirmed?
Applies to response 6: In my opinion, to assess the probability of mortality.
Applies to response 8: This is not the same. In the existing analyzes, the causes of deaths do not compete with each other, although in fact they do. Conventional time-to-event analysis ignores the probability of a competing event. Some of the statistical approaches, however, enable survival analysis in the presence of more than one endpoint. One of them, based on the Cox method, entails on estimation of type c event intensity, being termed the cause specific hazard. The risk of an event of interest is described in a discrete time setting as the number of cases who experienced the event of interest, divided by the number at risk at the time t. Some investigators do argue, however, that the method at issue is dubious, since it arbitrarily assumes an impossible to verify independence of events, as for each case an event of interest, a competing event, and censoring would have to occur in any order. Sensitivity analysis is therefore strongly advised, should there be any doubts as to the actual choice of optimum approach. The other statistical method is meant to calculate the hazard function of sub-distribution, followed by the estimation of regression model for this function. In this approach, a different hazard function is defined as the probability of endpoint, assuming that each case (patient) survived time t without the occurrence of event of interest, or experienced a competing event before time t.
Round 3
Reviewer 1 Report
Dear Authors,
I have not changed my belief in the validity of using the competing risk method in this study. Referring to the authors' arguments, I believe that in science the result cannot be based on the researcher's belief but should be verified with the use of an appropriate methodology. The desire to publish a manuscript should be no greater than a curiosity about the world .... but .... I am probably wrong sometimes.
Good luck